# Integrated Technology for Cereal Bran Valorization: Perspectives for a Sustainable Industrial Approach

**DOI:** 10.3390/antiox11112159

**Published:** 2022-10-31

**Authors:** Silvia Amalia Nemes, Lavinia Florina Călinoiu, Francisc Vasile Dulf, Anca Corina Fărcas, Dan Cristian Vodnar

**Affiliations:** 1Institute of Life Sciences, University of Agricultural Sciences and Veterinary Medicine Cluj-Napoca, Manastur 3-5, 400372 Cluj-Napoca, Romania; 2Faculty of Food Science and Technology, University of Agricultural Sciences and Veterinary Medicine Cluj-Napoca, Manastur 3-5, 400372 Cluj-Napoca, Romania; 3Faculty of Agriculture, University of Agricultural Sciences and Veterinary Medicine Cluj-Napoca, Manastur 3-5, 400372 Cluj-Napoca, Romania

**Keywords:** cereal bran, bioactive compounds, pretreatments, fermentation, health benefits

## Abstract

Current research focuses on improving the bioaccessibility of functional components bound to cereal bran cell walls. The main bioactive components in cereal bran that have major biological activities include phenolic acids, biopeptides, dietary fiber, and novel carbohydrates. Because of the bound form in which these bioactive compounds exist in the bran matrix, their bioaccessibility is limited. This paper aims to comprehensively analyze the functionality of an integrated technology comprising pretreatment techniques applied to bran substrate followed by fermentation bioprocesses to improve the bioaccessibility and bioavailability of the functional components. The integrated technology of specific physical, chemical, and biological pretreatments coupled with fermentation strategies applied to cereal bran previously-pretreated substrate provide a theoretical basis for the high-value utilization of cereal bran and the development of related functional foods and drugs.

## 1. Introduction

Cereals are the world’s most essential source of nutrients and energy for humans. According to the last report of the FAO (Food and Agriculture Organization of the United Nations), in 2018, the world produced 2.7 billion tonnes of grains, an increase of 18% from 2008. Furthermore, cereal grain output increased faster than the global population. The world’s most widely produced agricultural grains are wheat, rice, oat, corn, sorghum, barley, and rye. However, over one-third (1.3 billion tonnes) of the world’s food production for human consumption is wasted each year, particularly in the case of cereals, accounting for 30% (286 million tonnes) of total output [1].

At the same time, cereal by-products, such as cereal bran, are frequently devalued and utilized in animal feed, perceived as waste. However, the chemical composition of cereal brans is complex, and their various health benefits could be realized by incorporating them into the human daily diet. In addition to usual nutrients such as proteins, vitamins, minerals, and fats, cereal bran also contains a variety of bioactive compounds such as dietary fiber, phytosterols, biopeptides, novel carbohydrates, and polyphenols, such as phenolic acids and flavonoids [2]. The bioactive compounds have been demonstrated to exhibit a variety of biological functions, including anti-inflammatory, anticancer, antibacterial, and cardiometabolic protective effects [3]. In addition, they are linked to a reduced risk of chronic illnesses, such as chronic gut inflammation [4], obesity [5], cardiovascular disease [6], and cancer [7].

The mechanisms by which cereal bran confers health benefits are still to be fully established, in light of issues related to the digestibility, bioaccessibility, and bioavailability of compounds. Considering these limitations, a priority strategy targets the release of bound phenolic compounds, fiber, and minerals solubilization in situ before consumption, followed by their incorporation in fiber-rich functional food products. The food industry has been attempting to better exploit these by-products. Still, many improvements need to be made to pretreatment techniques to make the accessibility and solubility of the most interesting components valuable, as well as in the fortification into more nutritional and bioactive matrices to ensure sustainability.

Physical, chemical, enzymatic pretreatments, and bioprocesses may free the bound phytochemicals, leading to improved bioavailability and bioaccessibility for the bioactive compounds. The pretreatment method should be selected based on its techno-economic feasibility of integration into an efficient industrial process and its ability to minimize pollution levels in the environment. Therefore, modern and green pretreatments, such as ultrafine grinding, ultrasounds, microwave, steam explosion, and thermal and moisture pretreatments, have gained attention, being economically and environmentally attractive methods [8]. Thus, their optimization for specific matrices or integration for higher efficiency is in the early stage. Particular pretreatments can improve the digestibility of lignocellulosic biomass, with the expected main effects being hemicellulose dissolution and lignin structure modification, resulting in enhanced cellulose accessibility for further microbial enzymatic degradation and, ultimately, the production of new value-added products [9].

Microbial fermentation of bran substrates is an efficient, environmentally friendly bioprocess for increasing the release of bioactive compounds. The microorganisms involved in the bioprocess generate new compounds throughout fermentation. Agro-industrial residues are generally considered the best substrates for the fermentation processes [10,11]. Due to consumers’ desire for healthier diets, interest in naturally fortified foods, functional foods, and functional food additives is booming in the marketplace. Cereal bran, which includes wheat bran (WB), oat bran (OB), and rice bran (RB), is one of the most common, cheap, and desired ingredients for enhancing daily nutritional intake through food and supplement consumption [12].

In this respect, the present review aims to focus on the most promising new research progress in the development of an integrated system (Figure 1) consisting of physical, chemical, or biological pretreatments able to enhance the bioaccessibility of cereal bran phytochemicals before the bioprocesses (submerged fermentation and solid-state fermentation), resulting in a complex fortified grain matrix with valuable nutrients and bioactive compounds. The novelty of the proposed article is the approach of complete re-utilization of cereal bran via integrated pretreatments and in situ fortifications.

## 2. Intracellular Bioactive Phytochemicals of Cereal Bran

The chemical composition (Table 1) of non-wood agro-industrial residues, including cereal bran, which is composed of three layers, pericarp, testa, and aleurone, is dominated by macromolecular components, including cellulose (10.5–23.1%), hemicellulose (26.1–36.2%), and lignin (2.2–12.5%) [13]. Lower quantities of inorganic compounds and extractable components, such as phenolic compounds (0.7–2.7%), bioactive peptides, pectin, and lipids are also present (1.96–22%) [14]. Phytochemicals are bioactive compounds with reduced nutritional value found naturally in fruits, vegetables, and whole grains. Between 5000 and 25,000 distinct phytochemicals exist in these foods [15]. The nutritional compositions of crude WB, RB, and OB per 100 g are presented in Table 1.

### 2.1. Dietary Fibers

Non-starch polysaccharides, such as cellulose, hemicellulose, pentosans, gums, and glucans, found in cereal grain cell walls are a significant source of dietary fiber in the human diet. According to Wen Cheng and colleagues, dietary fiber is defined as ‘edible parts of plants or analogous carbohydrates resistant to digestion and absorption in the human small intestine with complete or partial fermentation in the large intestine’ [19]. Dietary fibers mostly consist of polysaccharides, oligosaccharides, and lignin fractions [19]. Dietary fibers are divided into soluble (primarily pentosans, pectin, gums, and mucilages) and insoluble (cellulose, some hemicellulose, and lignin) classes based on their water solubility [20]. The soluble fibers create a viscous polymer network in the presence of water, while the insoluble fibers remain impenetrable due to their structural rigidity. Cellulose is composed of cellobiose unbranched chains of d-anhydroglucose units β-(1,4)-glycosidic bonded embedded in a matrix of hemicellulose and lignin. Lignin is a component responsible for the rigidity and impermeability of cell walls and the inhibitory effect on cellulase activity. Lignin degradation through specific pretreatments is advantageous because it allows enzymes easier access to compounds of interest [21]. Furthermore, lignin provides a physical barrier to cellulose, and its presence leads to the formation of toxic compounds for fermentative microorganisms in hydrolysis. The cellulose and hemicellulose hydrolysis reaction produces fermentable sugars, which are required for the subsequent fermentation processes. The hard conditions in which cereal brans are pretreated generate sugar breakdown and dehydration of the substrate, resulting in the release of inhibitory substances such as (2-furaldehyde), 5-hydroxy-2-methylfurfural (5-HMF), and organic acids. The presence of β-glucosidases can reduce the inhibitory effect of lignin via the bonds that form between them, boosting the overall efficiency of the bioprocess [21].

Hemicellulose accounts for 20–35% of cereal biomass, is the second most prevalent cereal biomass, and has a significant value in various fields, including polymeric materials, foods, energy, and medicine. Compared to lignin, which is an amorphous polyphenol polymer, hemicellulose is a heterogeneous group of polysaccharides constituted by β-(1→4)-linked backbones of sugars, such as pentoses (xylose and arabinose) and hexoses (glucose, galactose, mannose, rhamnose, and glucuronic and galactouronic acids) [13]. The hemicelluloses in the bran cell wall are closely related to cellulose and lignin since their primary function is to act as a filler between the cellulose and lignin fractions. Furthermore, hemicellulose is more abundant than cellulose in cereal brans, making it the primary insoluble dietary fiber in cereal brans [13]. Arabinoxylans and glucans, particularly β-glucan, are the most studied hemicellulose groups in cereal brans.

Cereal bran pentosans, also known as arabinoxylans, are the predominant hemicellulose in the bran cell wall and influence the water distribution, starch retrogradation, and rheological properties of fermented grain-based food products [22]. Arabinoxylans consists of a linear chain of β-(1–4)-linked D-xylopyranosyl to which α-L-arabinofuranosyl substituents are attached [23]. The fraction of un-, mono- and di-substituted xylose and arabinose residues attached to the chain determines the variety of these molecules. The capacity of the arabinoxylan molecule to interact with other cell wall chemicals is also determined by its complexity, impacting the physicochemical and functional characteristics of the macromolecules [24]. Arabinose residues, which substitute the arabinoxylan molecule, are often composed of phenolic acids such as ferulic, caffeic, and p-coumaric acid. Arabinoxylan is a biomolecule with high antioxidant activity, excellent free radical scavenging capacity, reducing ability, and metal-chelating activity due to its structural complexity. Therefore, it can be well exploited in the food sector as a natural antioxidant [23]. 

Glucans are widespread D-glucopyranosyl polysaccharides present in cereals bran, including oat bran, wheat bran, and rice bran. The anomeric structure, the position and arrangement of the glycosidic linkages, the molecular size, and the type and degree of branching all differentiate glucans. Except for glucose residues connected by β-1, 3-linkages, known as β-glucans, most glucans have a reduced effect on the human immune system. According to the literature, glucans do not interact with the human immune system. The only interactions that can be observed are due to some glucose residues, found under the name of β-glucans [13]. Whereas β-(1,3) glycosidic connections restrict molecules from packing tightly, they give the β-glucan molecule an irregular shape, flexibility, and partial solubility in water [13,25]. Oat β-glucans provide beneficial health advantages, according to the European Food Safety Authority and the United States Food and Drug Administration, including immunomodulatory effects, the ability to reduce fasting blood cholesterol, and the ability to attenuate post-prandial glycemic reactions [25].

### 2.2. Phenolic Compounds

Phenolic acids are the primary compounds in cereal bran. There are three types of phenolic compounds and phenolic acids identified mainly in the three layers of bran (pericarp, testa, and aleurone): soluble free, soluble conjugated (glycosides), and insoluble bound states [26]. Most of the phenolic compounds from cereal bran are bonded in insoluble forms (Figure 2), except in the case of rice bran, which contains more free varieties of phenolic acids [2,15,27]. Bound phenolics may only be liberated by acid (H_2_SO_4_, HNO_3_, H_3_PO_4_, and HCl), alkaline (potassium hydroxide, sodium hydroxide), or enzymatic hydrolysis by applying commercial enzymes (endoxylanase, exoxylanase, or β-glucosidase) or via fermentation bioprocesses [28].

The two major classes of phenolic acids are hydroxybenzoic acids and hydroxycinnamic acids. Hydroxybenzoic acids are characterized by a C6-C1 structure and include gallic, vanillic, p-hydroxybenzoic, protocatechuic, syringic, and salicylic acids. In contrast, hydroxycinnamic acids, such as caffeic, ferulic, p-coumaric, chlorogenic, and synaptic acids, have a C6-C3 structure [29]. Ferulic acid accounts for most total polyphenols in wheat (90%) and oats (75%). Ferulic acid is commonly found in plants and results from phenylalanine and tyrosine metabolism. It is frequently seen as free ferulic acid, γ-oryzanol, monoesters containing ferulic acid, and several triterpene alcohols [30]. Significant concentrations of vanillic acid, syringic acid, salicylic acid, caffeic acid, and p-coumaric acid have also been detected in WB and OB; dihydroxybenzoic acids being the majority phenolic acids in OB [29]. Bran cell walls contain various complex structures, such as hydrolyzable tannins and lignins, that incorporate hydroxybenzoic and hydroxycinnamic acid derivatives [31]. The cell wall phenolics form ester linkages with structural carbohydrates, such as arabinoxylans, and proteins through their carboxylic group and ether linkages with lignin through their hydroxyl groups [15]. Ferulic acid and p-coumaric acid, the primary polyphenols in wheat bran, are found in the pericarp and aleurone of the grain. These chemicals are linked to the arabinoxylans in the grain layers through ester bonds strengthening the kernel’s structural integrity while also providing antibacterial, antifungal, and antioxidant protection. At the same time, some phenolic compounds are caught inside the bran matrix, forming hydrogen or hydrophobic bonds, significantly decreasing bioaccessibility and physiological effects. Moreover, the release and absorption of these compounds, in the form of digestion, encounter many difficulties [32].

Therefore, breaking the bran cell wall is critical for accelerating the release of phenolic compounds and enhancing phenolic bioavailability. The release of insoluble forms of phenolics can be increased by performing specific pretreatments on lignocellulosic cereal bran, such as physical, chemical, and biological/microbial pretreatment.

### 2.3. Bioactive Peptides

Proteins are well-known for their daily diet significance. Proteins, according to recent studies, are a rich source of biologically active peptides, such as Tyr-Ser-Lys, SSYYPFK, LQAFEPLR, and EFLLAGNNK [33,34,35]. Bioactive peptides are specific protein fragments formed by short-chain amino acids with health benefits in the human body beyond their essential nutritional value [36]. The positive effects of bioactive peptides include antihypertensive effects [37], antioxidant effects [38], antimicrobial effects [39], hypoglycemic effects [40], and immunomodulatory activity [41]. Compared to protein macromolecules, bioactive peptides have a simplified form and improved stability, while their immunogenicity is low or absent [40]. The presence of bioactive peptides has been noticed in many sources, such as cereals (oat, rice, and wheat), pseudocereals (amaranth, quinoa, and buckwheat), and legumes (soy, pea, and beans) [36]. Oats are grown and consumed worldwide, and their high protein content makes them an excellent source of bioactive peptides (12–24%). Globulins constitute the majority of bioactive protein and have the highest concentration of bioactive peptides, such as SSYYPFK from oat globulin hydrolysates, which can reduce systolic and diastolic blood pressure [34], and Tyr-Ser-Lys from RB powder, which had an antihypertensive effect [33,42]. Feng Wang and colleagues have identified 22 peptides of oat globulin from the tryptic hydrolysates [35]. Oat bioactive peptides provide the nutritional quality and positive impacts mentioned above due to their amino acid composition. Glutamic acid constitutes the majority of non-essential amino acids in oats, accounting for 25% of total non-essential amino acids, followed by aspartic acid, which accounts for 8% [33]. Lunasin is a novel 43-amino-acid biopeptide found originally in soybean cotyledon and later in cereals including rice and wheat bran. Lunasin has chemoprevention, anti-inflammatory, and antioxidant properties, and performs in vivo and in vitro obesity immunomodulation [5]. Tadashi Hatanaka and colleagues also show that bioactive peptides derived from rice and rice bran have anti-diabetic and antioxidant properties. Rice hydrolysates and rice bran were reported to exhibit modest antioxidant activity using the oxygen radical absorbance capacity assay with glutathione as the reference agent [43]. Ile-Pro (1.22 μg/mg), Met-Pro (0.23 μg/mg), Val-Pro (1.59 μg/mg), and Leu-Pro (1.94 μg/mg) dipeptides identified in the rice protein hydrolysate provide inhibitory activity against dipeptidylpeptidase-IV, a necessary preventive measure for type 2 diabetes [43]. 

### 2.4. Phytosterols

Phytosterols are bioactive molecules with a structure similar to cholesterol. Phytosterols are naturally found in plants whose central backbone is perhydrocyclopentanophenanthrene. In contrast to cholesterol, phytosterols are only absorbed through the daily diet via intestinal absorption because they cannot be synthesized endogenously in the human body [44]. Phytosterol-rich fractions could be used to increase the consumption of health-promoting compounds from natural sources in the cereal diet. The primary function of phytosterols in the human diet is to minimize the risk of hypercholesterolemia and cardiovascular disease. Clinical research has indicated that consuming 2 g of phytosterols per day is related to an 8–10% reduction in low-density lipoprotein cholesterol levels, lowering the risk of cardiovascular disease [45]. Some phytosterols are present in grains (rice, corn, wheat, and rye) as esters of ferulic acids, such as sterile ferulates. The distribution and composition of bioactive compounds in cereal grain varies within the whole grain, and phytosterols and steryl ferulates are usually concentrated in the bran layers. The most abundant phytosterols found in wheat are sitostanol, campesterol, β-sitosterol, and campestanol [45]. The results obtained in a study conducted by Nurmi Tanja and colleagues confirm that the phytosterols are concentrated in the WB fraction of wheat grain [46]. Moreover, the esterification of phytosterols with ferulic acid in WB can directly impact ferulic acid accumulation in the bran [46]. Rice, particularly RB and RB oil, are also important natural sources of phytosterols. Jiali Zhang and colleagues performed a study to investigate the relationships between the forms of RB and RB oil phytosterols and their bioactivities. Free sterols and sterol conjugates are the two primary structural forms of phytosterols. The structure of sterols is made up of four rings that form the cyclopentane-perhydrophenanthrene cycle that includes a hydroxyl group in the C3 position and a side chain of 8–10 carbon atoms attached to carbon 17 [47]. Free sterols are classified into 4-desmethyl sterols, 4α-monomethyl sterols, and 4, 4-dimethyl sterols, and conjugated sterols are fatty acid steryl esters, hydroxycinnamate sterol esters, steryl glycosides, and acyl steryl glycosides [48]. The conjugated form of sterile esters is characterized by the esterification of the hydroxyl group in the C3 position with a fatty acid. The conjugated form of sterile glycosides also includes the binding of a sugar molecule through a β-glycosidic bond to the C3 hydroxyl group in the sterol structure. Conjugated forms of acylsteryl glycosides include the esterification of the sugar fragment with the hydroxyl group in the C6 position. Moreover, sterol ferulates are a less frequent form of conjugated sterols, which is mostly present in grains [47]. Free sterols are cholesterol-like compounds that have received much interest for their health-promoting properties, such as reducing cholesterol or antioxidant effects. Different phytosterols have various bioactivities; for example, 4-desmethylsteryl ferulates have better antioxidant activity than 4-dimethylsteryl ferulates Field [48]. The structural form of phytosterols directly influences their bioactivity. Therefore, a deep understanding of structural models is required to evaluate phytosterols’ functionality.

## 3. Biological Activities in the Human Body

As stated above, whole grains are rich in polysaccharide compounds, lipids, proteins, and phytochemicals. Regular intake of whole grains or the addition of bran in the human daily diet has been confirmed in many studies to lower the risk of chronic diseases such as cancer, cardiovascular disease, and diabetes. The biological activities (antioxidant capacity, anti-inflammatory activity, cardiometabolic protective activity, anti-diabetes, anticancer, and prebiotic effects) of phytochemicals from WB, RB, and OB are discussed in this section, focusing on chronic illness prevention.

### 3.1. Antioxidant Capacity

The body’s antioxidant defense system can be unbalanced by an excess of free radicals that cause oxidative damage to intracellular biomacromolecules and induce oxidative effects on cells and tissues. Antioxidant substances protect molecules of high biological function, such as DNA, proteins, and lipids, from free radical damage, significantly reducing the risk of chronic diseases [49]. Polyphenols are the most researched compounds in cereal bran with remarkable antioxidant properties. The primary polyphenols having at least one aromatic ring in WB are hydroxycinnamic acid, ferulic acid, synaptic acid, and p-coumaric acid. P-hydroxybenzoic acid, vanillic acid, syringic acid, and gallic acid are the primary derivatives of hydroxybenzoic acid, another polyphenol in WB [50].

The antioxidant activity of phenolic acids in cereal bran can be linked to electron donation and the transfer of hydrogen atoms to free radicals that initiate oxidation processes in the human body responsible for the modification of cellular signaling processes, reduction of cellular destruction or cellular death, and the high incidence risk of chronic diseases [50]. The effectiveness of phenolic compounds in reaching target areas and performing any protective effect is determined by their bioavailability and bioaccessibility. Soluble polyphenols are absorbed into the systemic circulation in the stomach and small intestine, while bound polyphenols reach the colon without significant modification. Under the action of intestinal microbiota and specific enzymes, bound polyphenols can be released in the colon, producing bioavailable phenolic metabolites with potential health benefits, such as m-coumaric, hippuric acids, eriodictyol, 3′,4′,5′-trihydroxyphenylacetic acid, and 3′,5′-dihydroxyphenylacetic acid [29,51]. Chlorogenic acid, for example, was converted into caffeic acid during the first stage of microbial biotransformation (dehydroxylation, dehydrogenation, or ester hydrolysis), and the primary metabolites found included di- and mono-hydroxylated phenylpropionic acids, m-coumaric, and hippuric acid [51]. Recent studies have found that the concentration of phenolic compounds such as phenolic acids, flavonoids, and anthocyanins was positively associated with antioxidant activity [32,52]. Li et al. confirmed that particle size reduction increased the release of phenolic compounds of superfine WB, mainly ferulic acid and that the bioaccessibility of the phenolics increased to 65.51% compared to WB without physical treatment [32]. In addition, following the release of the bound phenolics from extruded rice bran by fungal fermentation, ORAC, and CAA antioxidant activity increased by 1.8 and 4.1, respectively [52]. Moreover, samples with the highest concentrations of phenolic acids, flavonoids, anthocyanins, and total phenolic content extracted with 65% (*v*/*v*) ethanol had the highest antioxidant activity based on DPPH radical scavenging activity and ferric reducing antioxidant power (FRAP) assays on two varieties of rice bran [53]. However, the health benefits of phenolic-rich substrates are directly influenced by the bioavailability of the phenolic acids.

The antioxidant potential of cereal fractions is frequently linked to the amount of ferulic acid present. Although bran contains the most ferulic acid, its bioavailability, and intestinal release are limited due to its structural arrangement in the bran matrix, where it is covalently bonded to indigestible cell wall polysaccharides [54]. Yang Li and colleagues investigated the phenolic release from WB during in vitro digestion and the influence of particle size on polyphenols release and antioxidant activity. By strongly decreasing the WB particle size followed by in vitro digestion, the primary phenolic chemicals produced, ferulic acid and p-coumaric acid, increased the capacity of superfine WB to scavenge 2,2-diphenyl-1-picrylhydrazyl radicals [32]. Oxygen radical absorption capacity and cellular antioxidant activity of free and bound phenolics increased significantly after the RB fermentation mediated by Aspergillus oryzae.

Moreover, in vitro gastrointestinal digestion of fungal fermented RB underlines the increasing phenolic content bioaccessibility during the fermentation bioprocess from 347.04 mg GAE/100 g to 567.16 mg GAE/100 g [52]. In white and dark purple RB samples, antioxidant activity was observed by ferric reducing antioxidant power and 2,2-diphenyl-1-picryl-hydrazyl-hydrate (DPPH). In human clinical trials, ferulic acid, vanillic acid, synaptic acid, and 3,4-dimethoxybenzoic acid bioavailability increased 2 to 3 times after consuming 300 g of bioprocessed bran for three days. Phenylpropionic acid and 3-hydroxyphenylpropionic acid were the primary metabolites of the bound phenolic compounds released in the colon [54].

The polysaccharides, bioactive peptides, and phenolic compounds in cereal bran have also been reported to have antioxidant activity. RB biopeptides demonstrated moderate antioxidant activity using the hydrophilic oxygen radical absorbance capacity assay. Inhibiting dipeptidylpeptidase-IV activity is one of the most significant type 2 diabetes prevention methods. Ile-Pro, Met-Pro, Val-Pro, and Leu-Pro are dipeptides with direct inhibitory activity against dipeptidylpeptidase-IV, which metabolizes the insulin-tropic hormone glucagon-like peptide-1, used to control postprandial hyperglycemia in type 2 diabetes, and to prevent and treat type 2 diabetes [43]. Furthermore, biopeptides derived from rice bran intensify the antioxidative responses in gingival epithelial cells, preventing periodontitis, a chronic inflammatory disease of the oral cavity which causes tooth loss in the absence of appropriate treatment [55]. The antioxidant activity of a new wheat bran polysaccharide constituted mostly of arabinose, xylose, mannose, glucose, and galactose fragments was analyzed by Xiao-Lan Shang and colleagues [56]. The investigated polysaccharide provides significant antioxidant activity according to DPPH, 2,2′-and-bis(3-ethylbenzothiazoline-6-sulfonic acid (ABTS), hydroxyl, and superoxide radical scavenging tests [56]. It is expected that introducing foods rich in antioxidant compounds, such as processed cereal bran, into the daily diet will help prevent the free radical-induced diseases that cause oxidative tissue degeneration.

### 3.2. Anti-Inflammatory Activity

Cytokines modulate the inflammatory process, which impacts the immune response through local and intracellular interactions. The two primary categories of cytokines identified are proinflammatory cytokines and anti-inflammatory cytokines. A clinical trial initiated by Nuria Mateo Anson and colleagues investigated the anti-inflammatory response after consuming processed wheat bran [54]. The proinflammatory cytokines were considerably reduced in blood samples collected from eight male subjects who consumed 300 g of bread containing processed bran for three days compared to the detected anti-inflammatory cytokines [54]. Statistically significant reduction in the inflammatory markers high-sensitivity C-reactive protein (32.66%) and tumor necrosis factor (13.06%) was obtained after ferulic acid diet supplementation for six weeks on hyperlipidemia subjects [57]. Obesity is a serious public health risk worldwide and is linked to low-grade chronic inflammation. Rice fiber components such as phytosterol pherulates and isoprenoids have been found to have anti-inflammatory properties, either directly on immune cells or indirectly through gut microbiota modulation [58]. A valuable clinical trial cross-examined the effects of rice bran on inflammatory factors and 12 weeks of hypocaloric diet in overweight and obese adults. Rice bran consumption combined with an energy-restricted diet has a significant impact on the levels of inflammatory factors due to the ability of fiber to reduce related inflammation factors such as inhibition of hyperglycemia and its effects on lipids, particularly LDL-cholesterol, and also due to the protection of small intestine health and consequently inflammation prevention [59]. Ewelina Kurtys and colleagues critically reviewed the modulation of intestinal microbiota following rice bran consumption, the impact on the central nervous system, and, indirectly, disorders connected to chronic neuroinflammation [58]. Multiple mental and neurodegenerative illnesses, including schizophrenia, depression, and Parkinson’s and Alzheimer’s diseases, have been linked to chronic neuroinflammation. Through the gut-brain axis, the fiber components in rice bran can lower inflammatory processes in the gastrointestinal tract and the brain [58]. A neuroinflammatory animal model investigated the effect of RB extract administration on memory and cognition after 21 days of treatment [60]. Peroxisome proliferator-activated receptor-gamma modulation was demonstrated to reduce the inflammatory response and improve cognitive dysfunction in neurodegenerative diseases. The study findings show that nuclear brain extracts from mice treated with RB extract increased the peroxisome proliferator-activated receptor-gamma [60]. Blood pressure control necessitates a specified focus on nutritional interventions involving a well-balanced and nutritious diet.

Dietary fiber has been demonstrated to lower blood pressure in hypertensive patients by increasing the relative abundance of SCFA-producing bacteria and beneficial bacteria in the microbiome. Bacteria from the genera *Bifidobacterium*, *Lactobacillus*, *Spirobacter*, and *Eubacteria* ferment the fibers that pass through the gut, creating significant SCFAs. *Trichosporium bacteria*, conversely, have anti-inflammatory properties and control bacterial disorders. The study’s findings revealed significant growth in bacteria of the genera *Bifidobacterium* and *Trichosporium* after a 3-month dietary intervention with OB [4]. Supplementing the daily diet with an adequate quantity of cereal bran, such as rice bran, oat bran, or wheat bran, can be used as nutritional prevention therapy for century illnesses. Moreover, cereal bran is a rich source of bioactive compounds used in clinical dietary intervention for patients with chronic disorders, such as non-communicable diseases, including type 2 diabetes, cancer, and cardiovascular disorders.

### 3.3. Cardiometabolic Protective Activity

Cardiovascular disease (CVD) affects the heart or blood arteries. It is typically related to fatty deposits inside the arteries (atherosclerosis) and an increased risk of coagulated blood. CVD is the leading cause of mortality for people all over the world. Inappropriate nutrition is the primary contributor to the development of CVDs. Oxidative stress, inflammation, hyperlipidemia, and hyperglycemia are major risk factors of inadequate nutrition.

Furthermore, there is a rising interest in consuming foods rich in bioactive phytochemicals that prevent future CVDs [61]. Cereal bran is one of the products with cardio-protective effects. The effects of a high-fat diet supplemented with 1% or 5% rice bran enzymatic extract on mice’s daily diet were researched by Cristina Perez-Ternero and colleagues [62]. According to the findings, consuming RB reduces oxidative stress and inflammation associated with atherosclerosis. The ferulic acid moiety of Oryzanol is the primary compound responsible for the observed effects [62]. Long-term dietary supplementation with 5% RB extracts for 23 weeks on wild-type mice diets provided similar findings. The study’s main conclusion was that RB extracts decrease cholesterol and avoid the development of atherosclerotic plaques [63].

A comprehensive meta-analysis observed the impact of replacing refined grains with whole grains on CVDs risk factors [6]. There were 25 randomized controlled trials, and 22 provided the criteria to be meta-analyzed. Whole-grain oats improved total cholesterol from SMD (standardized mean difference) = −0.54, 95% CI −0.95 to −0.12 and low-density lipoprotein cholesterol from SMD = −0.57, 95% CI −0.84 to −0.31, whole-grain rice improved triglycerides from SMD = 0.22, 95% CI −0.44 to −0.01, and whole grains, including wheat, oat, and rice, increased hemoglobin A1c from SMD = −0.33, 95% CI −0.61 to −0.04 and C-reactive protein from SMD = −0.22, 95% CI −0.44 to −0.00, according to the meta-analysis [6]. Therefore, when cereal bran is included in a regular and well-balanced diet, it provides considerable nutritional potential in preventing cardiometabolic illnesses.

### 3.4. Anti-Diabetes Activity

Diabetes is a non-communicable illness defined by a metabolic disorder in which the quantity of blood glucose becomes excessive due to a lack of insulin secretion. Diabetes is caused by various factors, including genetics and improper diet. Diabetes management primarily involves controlling blood glucose levels through proper diet, exercise, and hypoglycemia-prevention drugs [64]. Flour blends of rice bran and oat bran were examined with gluten-free multi-purpose product flour blends with blood glucose-reducing potential. The 70% wheat, 20% soy cake, 5% rice bran, and 5% oat bran mixture exceeded the other examined samples in terms of protein content (23.94 g/100 g), biological value (98.01%), and growth performance in rats [65]. Regarding the anti-diabetic potential of the experimental diets, the mix containing 5% rice bran and 5% oat bran had the strongest blood glucose-reducing action [65]. Timilehin David Oluwajuyitan and colleagues’ research produced similar results [66]. A blend with 5% rice bran and 5% oat bran had a low glycaemic index, glycaemic load, and high blood-glucose-lowering potential [66]. A meta-analysis of 15 randomized controlled studies conducted on human subjects from the United States, Canada, and Europe investigated the effect of the oatmeal-enhanced diet on the early stages of diabetes. The study’s findings indicate that consuming 3 mg of β-glucan from oats, corresponding to 60 g of oats, for at least eight weeks significantly lowered insulin, blood sugar, and glycosylated hemoglobin [64]. Therefore, personalized nutritional interventions are crucial in preventing and treating diabetes.

### 3.5. Anti-Cancer Activity

Cancer is one of the world’s most severe diseases, and it is one of the significant causes of mortality. Conventional cancer treatment procedures involve cytotoxic treatments such as radiation and chemotherapy, ignoring that both treatments frequently have significant toxic side effects. Natural ingredients are an excellent resource for developing novel and non-invasive anticancer therapies. The OB avenanthramides are a class of polyphenolic alkaloids that have both chemopreventive and chemotherapeutic properties. Avenanthramides’ potential therapeutic effect involves modulating many pathways, including stimulating apoptosis and senescence, reducing cell proliferation, and preventing epithelial-mesenchymal transition and metastatization [67]. After analyzing potential anticancer bioactivity against hepatocellular carcinoma cells, oat bran mixed with blackcurrant and blueberry powder showed enhanced anticancer benefits [7]. According to Hideyuki Nemoto and colleagues, oral intake of fermented brown rice and fermented rice bran with Aspergillus oryzae has a chemopreventive impact in rats, both on the growth of the tumor-associated inflammation and the subsequent development of metastasis [68]. The study found that mice fed with fermented rice bran and fermented brown rice had a reduced frequency of tumor cell development compared to mice fed a different diet. Furthermore, the metastatic capability of mice in the first category was significantly lower than that of mice on a regular diet who generated metastases to distant organs [68]. Additionally, the peptide fraction isolated from rice bran protein hydrolysate has anti-tumor solid activity against colon cancer cells [69].

### 3.6. Prebiotic Effects

Cereal bran is abundant in dietary fiber. A comprehensive analysis of wheat and rice bran dietary fiber revealed that they could modulate gut microbiota while also boosting the development of SCFA-producing bacteria [70]. The presence of fermentable soluble oligosaccharides, such as galactose-oligosaccharides, fructose-oligosaccharides, and xylooligosaccharides, contributes to bran’s prebiotic activity by modulating *Firmicutes*, *Verrucomicrobia*, *Enterobacteriaceae*, *Prevotella*, and *Bacteroides* gut bacteria [70]. An animal-designed study observed a decrease in the *Enterobacteriaceae*, *Streptococcaceae*, and *Enterococcaceae* mainly pathogen bacteria, and an increase in the *Lachnospiraceae* and *Ruminococcaceae* beneficial bacteria after eight weeks of feeding rice bran to the mice [71]. After 14 days of intervention, a randomized-controlled pilot clinical trial reported that 30 g/day of heat-stabilized rice bran resulted in a lower Firmicutes: Bacteroidetes ratio (2.7 down to 1.4) and increased SCFA (propionate and acetate) in feces [72]. Bacterial metabolic by-products, particularly SCFAs, are frequently regarded as essential mediators of gut-brain communication, and altered SCFA synthesis has been linked to allergies, asthma, cancers, autoimmune illnesses, metabolic diseases, and neurological diseases [73]. Arabinoxylan-oligosaccharide extracted from WB provided prebiotic effects in a double-blind, randomized, controlled, crossover trial. After three weeks of intervention, subjects who consumed 4.8 g/day of arabinoxylan as part of ready-to-eat cereal had increased bifidobacteria concentration in their fecal samples [74]. Therefore, cereal bran can be a natural source of prebiotic compounds that can be integrated as ingredients in food products and supplements to improve host gut health.

However, the health benefits of cereal bran phytochemicals are directly related to the number of phytochemicals available for absorption in the upper or lower section of the digestive system, known as bioaccessibility. It is critical to degrade the structural composition and liberate more bioactive compounds to increase bioaccessibility and modulate the functional value of cereal bran. Several pretreatments are widely required to support the release of bioactive from cereal bran and boost their bioavailability. Table 2 shows the main biological activities of cereal bran as well as the pretreatments done prior to the examinations.

## 4. Pretreatments’ Effect on Cereal Bran Phytochemicals

The aim of the biomass pretreatments is to fractionate the structure of the lignocellulose (Figure 3) in order to accelerate the chemical and biological processes of recovering the bioactive components connected to the cell wall. Untreated lignocellulosic materials are highly resistant due to cellulose crystallinity, matrix heterogeneity, limited surface accessibility, and lignin protection, as evidenced by their limited convertibility to fermentable sugars by enzymatic hydrolysis. At the same time, the pretreated biomass possesses characteristics that boost the accessibility and yield of cellulolytic enzymes in the conversion processes, such as higher porosity, a higher contact surface, and a low concentration of lignin [21]. Physical pretreatments consume a lot of energy; however, they are the most practical approach for treating biomass for additional bioprocessing.

Physical pretreatments (Figure 3a) include processes that reduce particle size significantly (e.g., ultrafine grinding) to increase conversion yield in bioprocesses such as fermentations [21,75]; processes that destroy and remove the protection provided by the rigid structures of hemicellulose and lignin (e.g., microwave pretreatments) [76]; processes that facilitate molecular interactions that cause mass transfer through cell membranes (e.g., ultrasound pretreatments) [77]; and forced separation (e.g., explosive steam pretreatments) [78]. Chemical pretreatments (Figure 3b) involve the removal of cellular components with high-molecular-mass, such as hemicellulose, pectin, and lignin, and the formation of a cellulose-based suspension for further enzymatic hydrolysis. Biological pretreatments (Figure 3c), on the other hand, are exceedingly specific but often time-consuming, with the primary objective of preparing the biomass before the application of a second bioprocessing step, such as microbial fermentation [21].

### 4.1. Physical Pretreatments

#### 4.1.1. Ultrafine Grinding

Cereal milling has been traditionally performed by physical procedures involving mechanical forces. Chemical changes, including changes in enzymatic sensitivity, occur due to particle size reduction operations and material changes. Ball milling is an efficient method of intensively reducing particles and macromolecules like cellulose into micro colloidal particles. The ball grinding method is applied for ultrafine grinding in wet and dry environments. Using balls of varying sizes and crushing the fibers in multiple directions helps increase the fibers’ exposed surface and their solubility level [75,79]. The reduction of the bran particle size supposes the growth of the internal surface of the cell wall, and the contact between the enzymes and the existing substrate is enhanced. The increased interactions between enzyme and substrate directly influence the release process of the bounded compounds. The increased access of enzymes to the exposed compounds as a result of the substantial particle size reduction boosts the extraction yield and, consequently, the bioaccessibility of the bioactive compounds. The grinding process is a physical method of modifying the physicochemical characteristics and improving the release of phenolic compounds by destroying the resistant structure of the bran [32]. The number of phenolic compounds released, particularly ferulic acid, increased significantly after in vitro digestion due to decreasing particle size. The content of free p-coumaric acid in superfine wheat bran (19.16 μm particle size) was almost five times higher (373%) than in coarse wheat bran (1110.39 μm particle size), medium wheat bran (235.68 μm particle size), and refined wheat bran (83.73 μm particle size). The aleurone layer of the superfine WB was entirely degraded, substantially increasing polyphenol extraction capacity and bioavailability. Following in vitro digestion of superfine WB, a significant release of ferulic acid was observed during intestinal digestion, and an improvement in the antioxidant capacity and inhibitory activity of the digestive enzyme of starch in the intestine [32].

Similar results were found when RB was exposed to intense milling. Compared to fine and coarsely milled powders, superfine RB powder had a higher content of bound flavonoids (30.98%). However, the concentration of extracted free flavonoids didn’t change the field considerably [80]. As a result of the intense decreasing particle pretreatments of cereal bran, the contact surface is enhanced. Therefore, the accessibility of phytochemicals contained in the fibrous matrix can be improved.

#### 4.1.2. Ultrasonic Pretreatments

The ultrasound principle is defined by mechanical vibrations between particles in an elastic medium generated by an ultrasonic wave [77,81]. The ultrasonic frequency is between 20 kHz and 10 MHz, higher than the human hearing range (16 Hz to 20 kHz) [77]. The energy released by the vibrations influences the interaction of the particles, resulting in the generation of thermal, cavitation, and mechanical effects [81]. Ultrasonic waves generate molecular interactions, which break cell wall structure, reduce particle size, and facilitate mass transfer across cell membranes. Cracks caused by cell wall disintegration enhance tissue permeability and increase the release of targeted compounds [77]. Chao Chen and colleagues investigated the influence of ultrasonic application and temperature on bound phenolics and β-glucan from defatted oat bran [82]. The pretreatments were followed by conventional extractions of the bioactive compounds and the determination of antioxidant capacity through the ORAC method and HPLC-DAD to identify and quantify the main phenolic compounds. Ultrasonic treatment enhanced β-glucan production by about 37%, and the phenolic content showed a higher yield following ultrasound application than maceration [82]. Maceration is the primary extraction method used for one to four days at various temperatures (from room temperature to boiling temperature) and solid-to-solvent ratios (from 1:1 to 1:20 g/mL) [83]. The ultrasound pretreatment method has been used to remove lignocellulose material and allow access to the nutrients by microorganisms for the growth and production of enzymes during rice bran solid-state fermentation (SSF) with Rhizopus oligosporus. Different ultrasonic parameters were studied and optimized, including ultrasound timing, cycling, liquid-to-solid ratio, and ultrasound amplitude conditions. Following the ultrasonic treatments, the required compounds for increased fungal growth were released, resulting in the liberation of the targeted bioactive compounds. The phytic acid concentration in the RB was reduced after ultrasonic pretreatment, while phytase production was increased after SSF [84]. Three minutes of high-intensity ultrasound (40 kHz, 11 W/cm^2^) pretreatment of fermented whey and oat beverages led to substantial Lactobacillus casei 431 growth, high antioxidative activity, and favorable consumer acceptability [85]. Using ultrasonic pretreatment on RB substrate supplemented with lingonberry pulp, followed by microbial fermentation, was an efficient strategy that enhanced the fermentation process, improved fermentation efficiency, and reduced fermentation time in Ruta Vaitkeviciene and colleagues’ study [86]. The quantity of soluble dietary fiber in RB increased by 17.5% after ultrasound pretreatment at 850 kHz and 160 W, for 20 min at 40 °C [86]. Therefore, ultrasonic pretreatments have an increasing potential to destabilize the rigidity of the cell wall to boost the bioavailability of the relevant compounds.

#### 4.1.3. Thermal and Moisture Pretreatments

The release of bioactive compounds from cereal bran and the increase in their bioavailability is also supported by temperature and moisture pretreatments [31]. Before the SSF, black RB was exposed to a physical pretreatment that included moisturizing to 30% and lifting in repose overnight, followed by autoclaving for sterilization. The major anthocyanin of RB, cyanidin-3-glucoside, was 99.58% reduced after physical pretreatment. Heat treatment causes anthocyanin’s circular structure to open, converting it to chalcone, a quickly degraded molecule. The reduction of cyanidin-3-glucoside due to increased protocatechuic acid content from 151.10 µg/g to 1055 µg/g by PP, possibly due to anthocyanin degradation during PP. The increasing of moisture to 30% activated some of the inherent enzymes of RB, which support the release of extractable phenolic acids during SSF [31]. Furthermore, the thermal process of 10 min at 80 °C boosted the total phenolic content of oat bran by 25.84% and wheat bran by 22.49%. Both WB (ferulic acid + 39.18%, vanillic acid + 95.68%, apigenin-glucoside + 71.96%, and p-coumaric acid + 71.91) and OB (avenanthramide 2c + 52.17% and dihydroxybenzoic acids + 38.55%) indicated a considerable ratio percentage increase in phenolic content following the thermal process [87].

#### 4.1.4. Steam Explosion Pretreatment

Steam explosion technology is a new high-efficiency modification approach. Steam explosion is a pretreatment procedure that involves exposing bran to saturated steam at high pressure for some time, followed by an immediate pressure reduction. Following steam explosion pretreatments, the crystalline structure of cellulose and hemicellulose is degraded due to cell rupture, mechanical fracture, and the structural rearrangement of biological macromolecules. Moreover, the conversion of insoluble fibers to soluble fibers occurs, while the solubility of soluble fibers is increased [88]. Steam explosion of wheat bran, followed by superfine grinding, increased water solubility index (211.75 mg/g), oil holding capacity (1.68 g/g), bile salts, and cholesterol adsorbing capacity (1.31 and 21.87 mg/g). Furthermore, total phenolic and flavonoid content increased significantly and indicated the highest DPPH radical scavenging activity (87.68%) [88]. The pretreatment efficacy is influenced by the type and physical accessibility of the tested samples. The samples are usually pretreated at a temperature of 160–260 °C, corresponding to a pressure of 0.69–4.83 MPa, for a short duration (30 s and 120 s) before being treated at atmospheric pressure. The principal impacts of steam explosion pretreatment are lignocellulosic structural deterioration, hemicellulose hydrolysis, and lignin fraction depolymerization. According to Liu and colleagues, steam explosion pretreatment (60 s at 185, 205, 215, and 225 °C, with corresponding pressure 1.02, 1.62, 2.05, and 2.45 MPa, and 215 °C for 30, 60, 90, and 120 s) successfully released wheat bran phenolic acids such as p-hydroxybenzoic, vanillic, syringic, p-coumaric, and ferulic acid [89]. Following pretreatment, the production and antioxidant capacity of soluble free and conjugated phenolic acids in wheat bran improved considerably [89]. The main economic benefits of steam explosion pretreatment are low costs, reduced energy consumption, high efficiency, and no pollution [88].

#### 4.1.5. Microwave Pretreatment

Pretreatments that include microwaves (MWs) on solid and liquid matrices induce the dipole to rotate, resulting in molecular friction and heat generation. Heat stress and high pressure in cellular tissues promote cellular structure breakdown and the release of targeted compounds [76]. Furthermore, cell disruption due to MW irradiation enhances the accessibility of polysaccharides in the cell wall and polymeric macromolecules are more fermentable during bacterial fermentation [76]. MW pretreatment before in vitro fecal fermentation enhanced the fermentability of previously extracted and isolated insoluble dietary fibers such as xylan A, xylan T, and arabinan, and the production of the SCFAs acetate, butyrate, and propionate [76]. MW irradiation was applied post-treatment after mixed fermentation of soybean by-products by Lactobacillus bulgaricus and Neurospora crassa, mixed in equal proportions as inoculum and fermented at 30 °C for two days with an inoculum size of 5.5%. The study’s findings indicate that SSF followed by MW irradiation at 600 W for 2.5 min significantly improved the soluble dietary fiber amount and composition from 9.55% to 20.92% while dispersing and stabilizing the fermented matrices [90]. Therefore, MW treatments may be used before and after bacterial fermentation, with beneficial results in both circumstances. MW treatments can be an attractive option from an economic perspective since processing time and energy consumption decrease, while also having favorable effects on bioactive compounds.

#### 4.1.6. Supercritical Pretreatment

Supercritical fluids can be used to pretreat lignocellulosic biomass under milder pretreatment conditions, resulting in higher sugar yields, reduced generation of fermentation inhibitors, and increased susceptibility to enzymatic hydrolysis while consuming fewer chemicals such as solvents, reagents, and catalysts [21]. Supercritical treatments use a technique in which supercritical fluids are introduced into the biomass, and bioactive compounds are separated or extracted based on solubility differences. Under regulated conditions, the compression and heating of a gas, such as carbon dioxide, change the gas’s physical properties, producing a supercritical fluid. The advantages of the supercritical fluid technique cover the elimination of toxic organic solvents, complete process automation, and sample oxidation prevention. At the same time, the technology’s limitations include the equipment’s high price, the inability to extract polar compounds, and high operating pressure, which requires a high energy consumption [78]. Carbon dioxide (CO_2_), along with ammonia, water, and hydrocarbons such as propane and butane, are among the most often used supercritical fluids [21,91]. CO_2_′s main advantage is the critical temperature of 31.1 °C, which allows for a wide range of applications in heat-sensitive bioactive compounds. Supercritical CO_2_ advantages include non-toxicity, non-flammability, reduced cost, minimal thermal energy consumption, and complete removal of the toxic residual solvent from the final extracts [91]. The supercritical fluid technique is performed as both pretreatment and extraction. The application of supercritical fluids as pretreatments improves the substrate’s accessibility to enzymatic hydrolysis by physical or chemical degradation of the lignocellulosic matrix. Furthermore, the fermentable sugar yield is significantly higher after pretreatment with supercritical fluids. In contrast, synthesizing fermentation inhibitory compounds, such as acetic acid, levulinic acid, formic acid, furfural, and 5-(hydroxymethyl) furfural, is minimal [21,91].

#### 4.1.7. Hydrothermal Pretreatment

Hydrothermal pretreatment is cost-effective because of its low energy consumption and the use of water as a solvent, which is inexpensive and does not cause environmental pollution [92,93]. Hydrothermal pretreatments can be subcritical or supercritical based on the critical points of water temperature and pressure, 374 °C and 22.1 MPa, respectively [92]. Changes in the physical characteristics of water around the critical temperature and pressure points allow the exact selection of the chemical reactions that occur. As a result, when water reaches the required temperature and pressure, the dielectric constant reduces, increasing the solubility of organic compounds. Secondly, when the temperature rises from 300 °C to 400 °C at a pressure of 25 MPa, the ionic product of water changes, and therefore the reaction mechanism changes [93]. Wheat bran was pretreated with supercritical water at a temperature of 400 °C, a pressure of 25 MPa, and a reaction rate of 0.19 s. Following pretreatment, high cellulose and hemicellulose hydrolysis efficiency was attained, with minimum degraded compound content. The essential part of preventing glucose deterioration was rigorous reaction time optimization [93].

### 4.2. Chemical Pretreatments

For many years, chemical pretreatments (CPs) have been used to increase the bioavailability of plant phytochemicals. However, because of high concentrations of ecologically hazardous composites, such as acids, alkalis, or oxidants, CPs have been observed to have unfavorable environmental consequences over time. Another result of applying CPs is using solid acids with a high degree of severity, such as H_2_SO_4_, HNO_3_, H_3_PO_4_, and HCl. They increase the synthesis of furfural and its derivatives, which are inhibitory compounds [94].

CPs are based on using acids and alkalis to solubilize bioactive compounds from cereal bran cell walls. The main factors in chemical pretreatments are the number of used reagents (acids and alkalis), temperature, reaction time, and the type and concentration of reagents. Appling acid-alkaline treatment (6.0 N HCl for the acid treatment and 6.0 N NaOH for alkaline treatment) on wheat bran, Huma Bader Ul Ain and colleagues brokedown the large polysaccharides into smaller oligosaccharide fractions and obtained increased amounts of soluble dietary fiber [20]. The main conclusions of the study were that the mode of action of consecutive acid-alkaline pretreatments opened the structure of the wheat cell wall and increased the fiber surface porosity, increasing the hydroxyl groups’ access to spread inside and perform the hydrolysis reaction during the subsequent alkaline treatment [20]. Lei Wang and colleagues obtained comparable results, which achieved acid treatment (H_2_SO_4_ and potassium hydroxide) on wheat bran to increase the concentration of soluble dietary fiber [95]. Following acid treatment, the structure of soluble fiber was considerably degraded, and the starch content was reduced. By increasing the concentration of H_2_SO_4_, the hydrolysis of hemicelluloses and amorphous cellulose was performed, increasing the crystallinity of the soluble fibers. The water retention, swelling, and oil-binding properties of the soluble fibers improved after the H_2_SO_4_ concentration was lowered below 2%. CPs influence the hypoglycemic properties of rice bran insoluble dietary fiber. Jing Qi and colleagues have found that the hypoglycemic effect of rice bran varies based on the pretreatment applied to rice bran [96]. After H_2_SO_4_ CPs on RB, the glucose fraction increased from 40.95% to 60.84%, increasing the CPs rigor, resulting in depolymerization of cellulose and noncellulosic components (e.g., starch). Different intensities of H_2_SO_4_ CPs decreased the xyloglucan fraction from RB hemicelluloses, resulting in a reduction in xylose, rhamnose, and fructose composition. As a result, applying acid in association with alkaline treatment will significantly boost rice bran fiber’s ability to absorb glucose while inhibiting α-amylase activity. Acid-alkaline treatments can increase the hypoglycemic properties of insoluble rice bran fibers, allowing them to be used as promising low-calorie functional components for fiber fortification [96].

CP applied to WB at the laboratory level costs about 4$ per kilogram of bran [97]. This cost can be recovered considerably due to the high-value biocomposites, such as dietary fiber and phenolic acids extracted from fermented wheat bran, a future-oriented perspective for cereal processors and growers.

### 4.3. Enzymatic Pretreatment

Most pretreatment procedures, including physical, chemical, and thermal procedures, require significant energy and volumes of chemical solvents. Furthermore, the rigorous environments they generate have adverse consequences on biomass, such as inhibitory by-products. Therefore, from these perspectives, biological pretreatments are more favorable since they do not require a significant consumption of energy and chemicals and do not generate inhibitory substances. However, industrial enzymes that break down lignocellulosic components, such as ligninolytic enzymes and cellulase, have a high unit price. Additionally, assuring the efficacy of pretreatments necessitates a high quantity of enzymes [94]. Conversely, enzymatic hydrolysis of lignocellulosic biomass is a safe and environmentally friendly pretreatment that may be successfully used in the food industry. Enzymatic hydrolysis involves the solubilization of insoluble components under heterogeneous environments by applying enzymatic blends of enzymes such as α-amylase, glucoamylase, protease, lysing enzyme, β-glucanase, xylanase, cellulase, feruloesterase, or pectinesterase. The hydrolysis of polysaccharide glycosidase linkages generates structural changes in the cereal bran matrices, lowering the rigidity and mechanical strength of cell walls [27]. Xue et al. applied β-endoxylanase and α-arabinofuranosidase enzymatic hydrolysis on WB. WB was first heated at 121 °C for 30 min and then incubated at 60 °C and pH 6.0 with the purified enzyme blend (β-endoxylanase and arabinofuranosidase) [98]. The quantity of xylooligosaccharides and phenolic acids, together with the antioxidant capacity, and water or oil retention capacity of WB rises after enzymatic processing. Furthermore, the results revealed that the degree of WB hydrolysis was enhanced, and a substantial synergetic effect was detected in the bran-containing dough and bread [98].

Due to their expanded application and industrial-scale efficiency, commercial enzyme pretreatments are among the most common treatments for phytochemical release. The fundamental limitation of this pretreatment is the high cost of enzymes. Therefore, further efficient methods for naturally releasing similar enzymes, such as fermentation processes, are required.

## 5. Cereal Bran Fermentation

Cereal bran bioprocessing is applied to enhance bran’s nutritional and technological value. The application of bioprocesses is designed to take advantage of the biological activity of cells and their parts to maximize the value of the bran matrix. Microbial fermentation of bran substrates is an efficient, environmentally friendly bioprocess for increasing the release of bioactive compounds. The microorganisms involved in the bioprocess generate new compounds throughout fermentation, the most frequent of which are microbial enzymes, biomass, and primary and secondary metabolites [99]. Among the microorganisms used in the fermentation of cereal bran can be mentioned molds or fungi, such as *Aspergillus* spp. [30,31,100], *Monascus* spp. [101], and *Rhizopus* spp. [102], bacteria such as *Lactobacillus* spp. [103], *Bifidobacterium* spp. and *Streptococcus* spp., and yeasts such as *Saccharomyces* spp. Microbial fermentation may be accomplished by two primary methods: solid-state fermentation (SSF) and submerged fermentation (SmF) (Figure 4). 

Solid-state fermentation (SSF) is a food processing method in which microorganisms grow and multiply without free water. It has been used for thousands of years and recently has received increased interest because of the compositional changes it may generate in agro-industrial wastes, particularly their bioactive components. Microorganism-produced enzymes and endogenous enzymes implicated in SSF, and those involved in SSF can liberate bound phenolic compounds and bioconvert the phenolic metabolites, enhancing cereal bran bioavailability and biological activities (Table 3) [100,103]. Cereal bran represents a matrix with low free water content.

To maximize the efficiency of the SSF bioprocess, several parameters must be optimized, including the fermentation microorganisms selection, substrate composition to ensure the carbon source required for microorganism growth and multiplication, potential pretreatments to the matrix, and bioprocess temperature, aeration, pH, and moisture [28]. Călinoiu and colleagues evaluated the potential of SSF to enhance the phenolic content and antioxidant activity of WB and OB using the Saccharomyces cerevisiae yeast [10]. The maximum concentration of total phenolic compounds was reached after the first three days of fermentation in the case of WB, and the fourth day of fermentation in the case of OB. WB recorded a 56.6% increase in ferulic acid, 259.3% vanillinic acid, and 161.2% dihydroxybenzoic acids, whereas OB fermentation increased the ferulic acid by 21.2%, and the avenanthramide by 48.5% [10].

Similarly, after RB SSF with Rhizopus oryzae, the biological activities (antioxidant activity, anti-inflammatory activity, and ferric-reducing power) and main bioactive compounds (phenolics, flavonoids, carotenoids, and anthocyanin) significantly increased [102]. Similarly, following the SSF on WB substrate, the soluble phenolic content of wheat bran was nearly triplicated compared to the raw bran [104]. The significant increase in total phenolic compound concentration is primarily due to acid hydrolysis during fermentation, which accelerates the breakdown of the bonds that link the phenolic compounds to the cell wall polysaccharides [103]. At the initiation of SSF, microorganisms come into contact with the bran cell wall, mainly composed of cellulose, hemicellulose, and lignin. Therefore, during the first fermentation stage, the microorganisms naturally produce a large number of enzymes to break down the cell wall and get accessibility to the lignocellulosic compounds [28]. According to Bei and colleagues, xylanase activity produced by *Monascus anka* throughout SSF of OB was highly related to phenolic content. Xylan is a significant component of the cell wall. The high activity of xylanase during the first days of fermentation resulted in the breakdown of OB cell walls and the consequent release of phenolic compounds [101]. Therefore, in addition to phytochemical compound synthesis and extraction efficiency enhancement, SSF has excellent potential to be applied to bran biomass to produce microbial enzymes and develop ingredients with added value for food, pharmaceutical, and cosmetic applications.

Submerged fermentation is a bioprocess wherein microorganisms grow inside a fluid medium rich in essential nutrients and with an appropriate oxygen concentration for microorganism growth and multiplication [99]. The SmF bioprocess, like the SSF bioprocess, involves the release of bound bioactive compounds from the bran matrix via microorganisms involved in fermentation and the bioconversion and production of additional high-value compounds as secondary metabolites. Ferulic acid was liberated and biotransformed by lactic acid bacteria into phenolic derivatives such as 4-ethylphenol, vanillin, vanillic acid, and vanillyl alcohol after 24 h of rice bran SmF. Ferulic acid esterase and ferulic acid decarboxylase were the enzymes that increased the efficiency of the bioprocess. The study’s key finding was the potential of lactic acid bacteria to synthesize in situ biovanillin through Smf of agro-industrial lignocellulosic waste, such as cereal bran [105]. Nowadays, synthetic vanillin (4-hydroxy–3-methoxy benzaldehyde) dominates the vanilla-flavoring industry in the food, beverage, pharmaceutical, cosmetic, and tobacco areas. Therefore, SmF and SSF biotechnological processes using low-cost substrates like agro-industrial by-products are likely to develop new vanilla synthesis pathways via strain metabolism [106]. The primary advantages of SSF over SmF are high yield, enhanced product stability, low protein degradation, and minimal contamination risk. Another advantage is the reduced production cost. According to economic analysis, the production cost of cellulase enzyme via SSF (15.67 USD/kg) is three times lower than obtaining a similar enzyme using SmF (40.36 USD/kg), but also advantageous because the enzyme’s marketing price is 90 USD/kg [104]. SmF, on the other hand, provides many advantages that support its use in many applications, such as efficient control of fermentation parameters such as temperature, humidity, pH monitoring, and optimal aeration due to continuous homogenization through the fermentation. Furthermore, the SmF bioprocess can minimize the danger of fungal hyphae drying up when fermentation occurs in molds [99].

**Table 3 antioxidants-11-02159-t003:** SSF and SmF applied to cereal bran.

Fermentation Method	Inoculum	Substrate	Fermentation Condition	Results	References
SSF	*Enterococcus faecalis* M2	WB	Inoculation rate: 10%; Moisture content: 60%; Time: 36 h; Temperature: 37 °C.	Soluble dietary fiber ↑ Phenols ↑ Flavonoids ↑ Alkylresorcinols ↑ Free amino acid ↑ Protein ↓ Phytic acid ↓ Antioxidant capacity ↑	[103]
	*Bacillus* sp. TMF–2		Inoculation: 0.5 mL of bacterial suspension; Solid-to-liquid ratio: 1:1; Time: 11 days; Temperature: 30 °C.	Soluble phenolic content ↑ Antioxidant capacity ↑ Free radical scavenging rate ↑ Activity of hydrolytic enzymes (amylase, cellulase, pectinase, mannanase, protease, and phytase) ↑ Inorganic phosphorus ↑ Phytic acid ↓	[104]
	*Aspergillus strains*: *A. brasiliensis*, *A. awamori*, and *A. sojae.*	RB	Moisture: 1:1 (*w*/*w*); Fungal spores: 1% (*v*/*w*); Time: 8 days for *A. brasiliensis* and 14 days for *A. awamori* and *A. sojae;* Temperature: 25 °C.	Radical scavenging activity ↑ Tyrosinase inhibitory activity ↑ Lactase inhibitory activity ↑ Kojic acid ↑ Free phenolic acids ↑ Total flavonoid content ↑	[100]
	*Rhizopus oryzae*		Moisture: 50%; Spore concentration: 4 × 10^6^ spores/g bran; Time: 96 h; Temperature: 30 °C.	Total phenolic ↑ Total flavonoid ↑ Total carotenoid ↑ Total anthocyanin ↑ Antioxidant capacity ↑ Ferric reducing power ↑ Anti-inflammatory properties ↑ Anti-diabetic properties ↓ Radical scavenging ability ↓	[102]
	*Aspergillus awamori* and *Aspergillus oryzae*		Moisture: 30%; Inoculation: 1 mL of the spore suspension; Time: 5 days; Temperature: 30 °C.	Total phenolic content ↑ Protocatechuic acid ↑ Ferulic acid ↑ Radical scavenging activity ↑ Tyrosinase inhibitory activity ↑	[31]
	*Saccharomyces cerevisiae*	OB	Moisture: 45% (*w*/*w*); Inoculation: 5 mL of yeast suspensions (10^7^ CFU/mL) per 100 g of dry weight; Time: 6 days; Temperature: 30 °C; Static conditions.	DPPH radical activity ↑ Total phenolic content ↑ Avenanthramides ↑ Ferulic acid ↑ Protocatechuic acid ↑ Caffeic acid ↑ Vanillic acid ↑	[10]
	*Monascus anka*		Moisture: 60% (*w*/*w*); Inoculation: 0.1 mL of spore suspension per 1 g dry oats; Time: 14 days; Temperature: 30 °C	Ferulic acid ↑ Vanillic acid ↑ α-amylase activity ↑ Xylanase activity ↓ Total cellulase activity ↓ β-glucosidase activity ↓	[101]
SmF	3-member consortium of *Bacillus subtilis*, *Bacillus coagulans*, *Bacillus cereus*	WB	Wheat bran 2% (*w*/*v*); Time: 7 days; Temperature: 30 ± 2 °C; Agitation speed: 140 rpm.	Cellulases activities ↑ Digestibility of solid substrates ↑ Cellulose bioconversion ↑ Lignocellulose degradation ↑ β-glucosidase ↓	[107]
	*Aspergillus phoenicis (Aspergillus saitoi)*		Wheat bran: 1% (*w*/*v*); Temperature: 40 °C; pH: 6; Inoculation: 10^5^ spores/mL.	Production of thermo-toleran mycelial β-D-fructofuranosidase and raffinose;	[108]
	*Pediococcus acidilactici*	RB	Inoculation: 3% (*v*/*v*); Time: 24 h; Temperature 37 °C; pH: 5.6	Bioconversion of ferulic acid into phenolic derivatives such as 4-ethylphenol, vanillin, vanillic acid, and vanillyl alcohol.	[105]
	*Saccharomyces cerevisiae*		Inoculation: 25 mL (1 × 10^8^ cells/mL); Constant aeration; Agitation speed: 150 rpm; Time: 24 h; Temperature: 35–45 °C; pH: 3.5–4.5.	Antioxidant properties ↑ Cyanidin-3-glucoside and peonidin-3-glucoside were bioconverted to cyanidin and peonidin Bioactivity ↑	[109]
	*Bacillus mojavensis*	OB	Inoculation: 0.5%, 1%, and 2% (*w*/*v*); pH: 8.0 Time: 120 h; Temperature: 37 °C; Agitation speed: 180 rpm.	Xylanase yield of about 249.308 IU/mL	[110]

## 6. Conclusions and Future Perspectives

The fortification and re-use of cereal milling industry by-products to develop new functional foods is a research direction of great interest and practicality from the perspective of food–health relations, as well as from the environment protection and waste management perspectives. Due to the high concentration of bioactive compounds in cereal bran that have valuable biological functions, such as antioxidant activity, anti-inflammatory activity, cardiometabolic protective activity, anti-diabetes activity, anti-cancer activity, and prebiotic effects, cereal bran phytochemicals have lately attracted a lot of interest from the food and medical industries. The proportion of phytochemicals in cereal bran that are bioaccessible, or available for absorption in the digestive system, is closely correlated with the health benefits of cereal bran. SSF and SmF bioprocesses significantly increase the production of phytochemicals, particularly antioxidant phytochemicals, such as phenolic compounds, through the secondary metabolic pathway of fermentation microorganisms or following extracellular enzymatic activity after the release from the substrate matrix. The accessibility of microorganisms to bran phytochemical compounds contributes to the efficacy of fermentative bioprocesses. The rigid structure of cell wall polysaccharides such as lignin, hemicellulose, and cellulose protects the high-value bioactive compounds. Specific pretreatments are required to break the lignocellulosic structure and make cereal polysaccharides accessible to hydrolytic enzymes, allowing the fermentable sugars to be freed. Moreover, the porosity and specific surface of the insoluble fiber of the bran can be enhanced by applying chemical, physical, and biological pretreatments, resulting in increased bioavailability of cell walls phytochemicals. Pretreatments that are modern and environmentally friendly, such as steam explosion, ultrafine grinding, and ultrasound and microwave pretreatments, have gained attention because they are both efficient and cost-effective. As a result, more work should be done on optimizing these pretreatments for particular matrices or integrating them in the early stage of the bioprocesses for greater efficiency. However, to scale the pretreatment’s applicability, more research is required to evaluate these integrated methods’ environmental effects, cost efficiency, and energy balance.

## Figures and Tables

**Figure 1 antioxidants-11-02159-f001:**
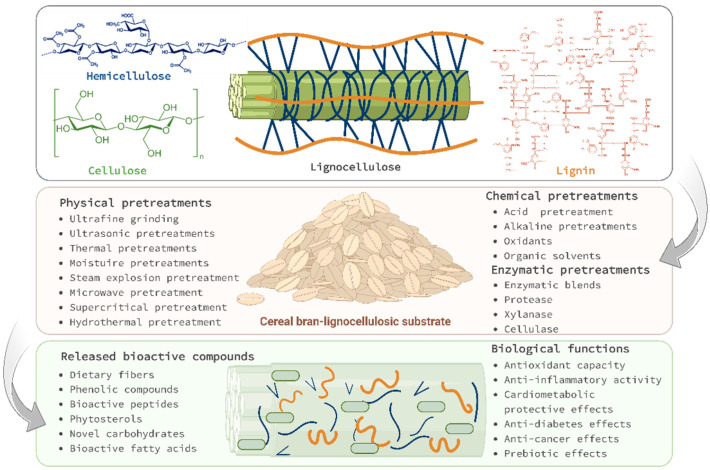
Integrated system to enhance the bioaccessibility of cereal bran phytochemicals.

**Figure 2 antioxidants-11-02159-f002:**
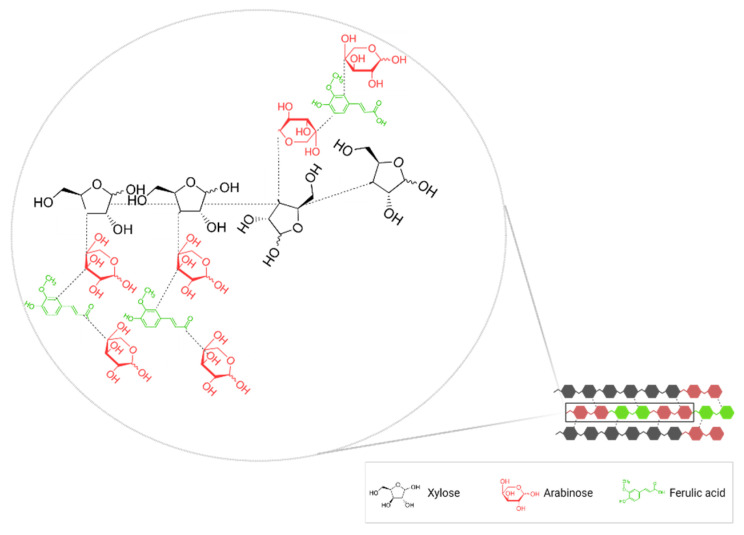
Cell-wall bound ferulic acid in cereal bran.

**Figure 3 antioxidants-11-02159-f003:**
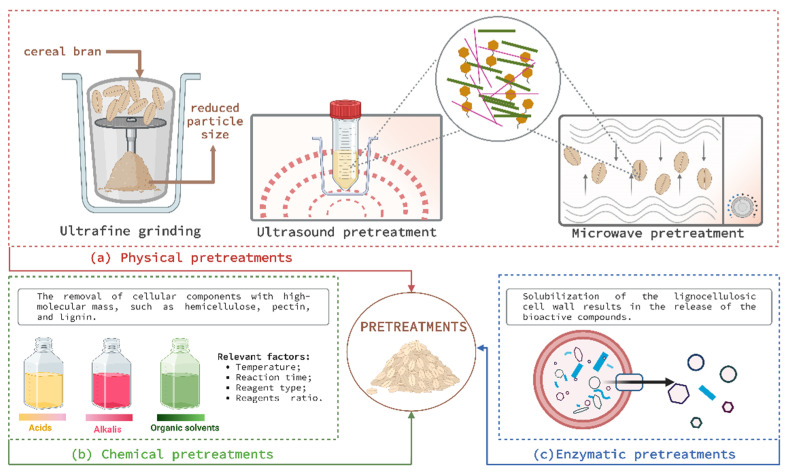
Key pretreatments for cereal bran processing.

**Figure 4 antioxidants-11-02159-f004:**
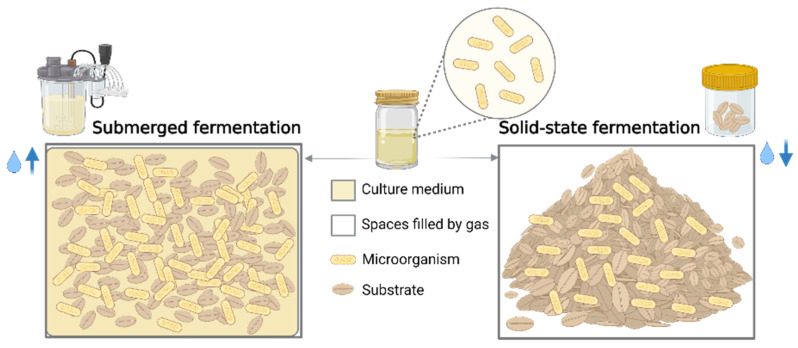
Solid-state fermentation and submerged fermentation.

**Table 1 antioxidants-11-02159-t001:** Cereal bran nutritional composition.

Nutrients	Wheat Bran [16]	Rice Bran [17]	Oat Bran [18]
	Amount (g/100 g)	Amount (g/100 g)	Amount (g/100 g)
Water	9.89	6.13	6.55
Energy	216 kcal	316 kcal	246
Protein	15.6	13.4	17.3
Total lipids	4.25	20.8	7.03
Polyunsaturated fatty acids	2.21	7.46	2.77
Monounsaturated fatty acids	0.63	7.55	2.38
Ash	5.79	9.98	2.89
Carbohydrates	64.5	49.7	66.2
Fibers	42.8	21	15.4
Total sugars	0.41	0.9	1.45

**Table 2 antioxidants-11-02159-t002:** Biological activities of wheat bran, rice bran, and oat bran.

Source	Study	Pretreatments	Biological Activity	Key Findings	References
Wheat bran	in vitro	Ultrafine grinding: Coarse wheat bran Medium wheat bran Fine wheat bran Superfine wheat bran	Antioxidant activity; Digestive enzymes inhibitory activities	*p*-coumaric acid content in superfine WB was five times higher; Phenolic compounds bioaccessibility was increased by 65.51%; Starch digestibility was reduced.	[32]
	in vivo	Enzyme pretreatment (xylanase, cellulose, β-glucanase, and feruloyl-esterase) Yeast fermentation (Baker’s Yeast)	Antioxidant capacity; Anti-inflammatory properties	Phenylpropionic acid and 3-hydroxyphenylpropionic acid were the main colonic metabolites identified; The pro-:anti-inflammatory cytokines ratio was significantly lower after the consumption of 300 g of bioprocessed bran-based bread for 3 days.	[54]
	in vitro	Grinding	Antioxidant activity	The extracted xylose, mannose, glucose, and galactose exhibited remarkable antioxidant activities (DPPH, ABTS, hydroxyl, and superoxide radical scavenging tests)	[56]
	in vivo	-	Prebiotic effect	Arabino-xylan-oligosaccharide provided prebiotic properties increasing fecal bifidobacteria and postprandial ferulic acid concentrations.	[74]
Rice bran	in vitro	Extrusion	Antioxidant activity;	Extractability of the bound phenolics was increased; Increased free bound and total phenolic content by 23.0%, 50.7%, and 36.3%; No effect was observed on CAA antioxidant activity; Bioaccessibile phenolics increased by 40.5%	[52]
	in vitro	Fungal fermentation	Antioxidant activity;	Extractability of the bound phenolics was increased; Increased free bound and total phenolic content by 99.4%, 40%, and 71.6%; ORAC antioxidant activity increased 1.8-fold; CAA antioxidant activity increased 4.1-fold; Bioaccessibile phenolics increased by 64.5%	[52]
	in vitro	Enzymatic pretreatment (protease from *Aspergillus oryzae)*	Anti-diabetic activity; Antioxidant activity;	Dipeptides Ile-Pro, Met-Pro, Val-Pro, and Leu-Pro had shown inhibitory activity against DPP-IV (dipeptidylpeptidase-IV); No inhibitory activity against human maltase–glucoamylase was observed.	[43]
	in vivo (animal study-mouse model)	Maceration in ethanol	Neuroinflammatory responses (memory and cognitive performance)	Spatial working, reference memory, and non-spatial recognition memory were improved	[60]
	in vivo (animal study-mouse model)	Enzymatic hydrolysis	Antioxidant capacity; Anti-inflammatory properties	Oxidative stress was reduced; Ferulic acid and γ-oryzanol reduced pro-inflammatory monocyte phenotype; Atherosclerosis-related oxidative stress and inflammation were reduced.	[62]
	in vivo (animal study-mouse model)	Fermentation with *Aspergillus oryzae*	Anti-inflammatory activity; Anti-mutagenic effects	Fermented brown rice and rice bran provided chemopreventive effectiveness against inflammation-related carcinogenesis	[68]
Oat bran	in vivo (animal study-mouse model)	Grinding	Antidiabetic activity; Carbohydrate hydrolyzing inhibitory activity; Antihyperglycaemia activity.	The samples with 70% wheat, 20% soy cake, 5% rice bran, and 5% oat bran provided low glycaemic index, high carbohydrate hydrolyzing enzyme inhibitory potential, and blood glucose lowering potential.	[65]
	in vivo	Maceration	Antidiabetic activity	The samples with plantain 60%, soy cake 30%, rice bran 5%, and oat bran 5% had the highest blood glucose-reducing activity.	[66]

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
