# Peer review of "Integrated Technology for Cereal Bran Valorization: Perspectives for a Sustainable Industrial Approach"

_antioxidants, 2022, doi:10.3390/antiox11112159_

Round 1
Reviewer 1 Report
Dear author,
The paper includes very useful information as regards cereal bran valorization, including bioactive compounds, their effect, pretreatments, etc. However, I would recommend to consider these aspects:
Title. I think the title does not cover the information included in the paper. I would recomend to change it. Integrated technology???
Page 2, line 88. Includes the frequent % range of cellulose, hemicellulose, and lignin
Line 89-90. Which % represents these compounds?
Line 97. Reference of Wen Cheng
Line 102. Include behavior of soluble and insoluble fibre. For example, soluble fibre absorbs water in order to form a viscous polymer network.
Line 103. Which percentage do they represent?
Line 106. Percentage of lignin
Line 107. Explain this inhibitory effect
Line 108. Explain which enzymes and their role
Line 110. You mentioned the process of hydrolysis but it is not described which is the purpose of this process
Line 116. Which sugars mainly?
Line 119. Remove “however”
Line 139. Explain “reduced effect on the human immune system”
Line 148. Maybe the layers of bran should have been mentioned before talking the different components, no just when you start talking about phenolic compounds
Line 152-153. Explain the most frequent acids, alkali and enzymes used with this purpose
Line 158-160. English revision required
Line 172-174. English revision required
Line 206-208. Revise the phase and complete the information
Line 208. Where are these peptides (Ile-Pro, Met-Pro, Val-Pro, and Leu-Pro RB) found?
Line 210. Where are these oligopeptides founded? In which proportion?
Line 228-229. Include the reference of Nurmi Tanja
Line 234. Explain particular structure of Free sterols and sterol conjugates to explain different properties
Line 243. Mentioned that this information has already given
Line 243. Include “of” in the excess “of” free radicals
Line 272. Which metabolites
Line 299. Explain the effect of fungal growth on the substrate
Line 321. Reference of Xiao-Lan Shang is missing
Line 333-334. Reference is missing
Line 349-352. Reference is missing
Line 356-357. Reference is missing
Line 361. Explain the role of peroxisome proliferator-activated receptor-gamma
Line 369. Remove “.”
Line 365. Which is the role of the short-chain fatty acid-producing bacteria (SCFA)
Line 364-366. Phrase needs revision
Line 372. Century illnesses?
Line 386-387. Reference is missing
Line 393. Reference is missing
Line 410-411. Reference is missing
Line 412-414. Explain more the information. It is not clear
Line 416-417. Reference is missing
Line 441. Aspergillus oryzae is italic
Line 439-442. Reference is missing
Line 457-460. Explain if these microorganisms are pathogens or beneficials.
Line 462. Explain Firmicutes: Bacteroidetes ratio
Line 462. Which is the benefit of the increased of SCFA?
Line 473-474. It is mentioned that “Several pretreatments are widely required to support the release of bioactive from cereal bran and boost their bioavailability”. In all the examples given in section 3.6, where the biomass previously treated to evaluate the healthy effect?
Line 488. Which is the role of these enzymes?
Line 517-518. Reference is missing
Line 523. Explain maceration
Line 523. It is better to say that “has been used”
Line 523-526. Reference is missing
Line 538. Reference is missing
Line 545. Bioactive “compounds”
Line 554-556. Reference is missing
Line 555. Explain this thermal pretreatment
Line 556-557. Do you mean that fungal pretreatment was better than thermal?
Line 573. Include times frequently used
LInes 576-578. Reference is missing
Line 592-594. Give more information
Line 624-638. Hydrothermal is a pretreatment method or a hydrolysis method
Line 650. And also the type and the concentration
Line 654. Reference is missing
Line 651. Which specific reagents were used in this study?
Line 659. Reference is missing
Line 661-665. Phrase needs revision
Line 667. Reference is missing
Line 671-673. Explain more
Line 686. Also it should be considered the environmental impact
Line 692. Other enzymes are also important as cellulases
Line 693. Include the typical commercial enzyme cocktails used for this purpose
Line 696. Reference is missing
Line 698. Include incubation temperatura
Line 719. Microorganisms in italic
Section 5. Which are the most frequent microorganisms for this purpose
Line 770. Reference is missing
Table 1. Uniformity in units like, for example, moisture
Title os section 5 should be modified as the enzymatic pretreatment could be also considered a biological process
Conclusion section should be modified according to information included in the paper. For example, biological activities of bioactive compounds represent an important part of the paper and they are not practically mentioned in the conclusions.
Reviewer 2 Report
The submitted review deals with improving the bioaccessibility of functional components bound to cereal bran cell walls. The study fully reviewed the intracellular bioactive phytochemicals of cereal bran, its biological activities in the human body, pretreatments effect on cereal bran phytochemicals, and the cereal bran bioprocessing. The study is relevant in the field of the journal, and the writing is ok. We suggest that this manuscript could be published after a major reversion. The specific points are as follow:
1. One of the most important point for a review is to summarize the latest research progress. In this manuscript, much effort has been applied to the introduction of basic contents, especially in the Section 2. Thus, it is advised to added the latest work and new academic viewpoints for the integrated technology for cereal bran valorization.
2. It is advised to show the key pretreatments in a intuitive illustration, and also for the other related part in this manuscript.
Reviewer 3 Report
Review Report
The review entitled “Integrated technology for cereal bran valorization: Perspectives for a sustainable industrial approach” authored by Silvia Amalia Nemes et al reports the Intracellular bioactive phytochemicals of cereal bran providing detailed information on their structure, functionality, and bioactivity to further present the effect of natural pretreatments and of the bioprocessing on bran matrixes with a final goal of obtaining higher levels of bioavailable phytochemicals.
The review study is very interesting since it collects material demonstrating that the potential bioactivity of cereal bran has been underutilized and recommends strategies to fully use it.
However, I do have some concerns and suggestions to improve the manuscript before it can be accepted for publication:
Minor comments
1. Authors should correct the numbering of the Conclusion section from 5. to 6.
Major comments
1. I would suggest that the authors accompany the text in sections 3 and 4 with informative tables as they have done for section 5 (Cereal bran bioprocessing)
2. Authors should discuss in a critical way the so far data and point out research that must be done in the future on this topic.
Round 2
Reviewer 1 Report
Dear author,
The paper has been significantly improved after your corrections. The only thing that should be changed is the subscript of line 745.
Reviewer 2 Report
The manuscript is revised appropriately and the writting is well improved.